# Spatiotemporal patterns of tuberculosis in urban slums and urban–rural transition zones: evidence from Tétouan, Morocco, 2019–2023

Ayoub Ez-Zari[1]*, Zaida Herrador[2,3], Laila Farouk[1], Nadya Mezzoug[4], Abdnnor Boulaich[5], Khalid Bouti[6,7], Zakaria Mennane[1], Noureddine Elmtili[1]

1 Department of Biology, Laboratory of Biology and Health, Food Science and Health Research Team (UAE/U16FS), Faculty of Sciences, Abdelmalek Essaâdi University, Tétouan, Morocco, 2 National Center for Epidemiology, Instituto de Salud Carlos III, Madrid, Spain, 3 CIBER Epidemiología y Salud Pública (CIBERESP), Madrid, Spain, 4 Department of Biology, Laboratory of Applied Chemistry, Microbiology and Biotechnologies, Faculty of Sciences, Abdelmalek Essaâdi University, Tétouan, Morocco, 5 Department of the Health Establishments Network, Ministry of Health and Social Protection, Tétouan, Morocco, 6 Laboratory of Life and Health Sciences, Faculty of Medicine and Pharmacy of Tangier, Abdelmalek Essaâdi University, Tangier, Morocco, 7 Department of Pulmonology, Mohammed VI University Hospital, Tangier, Morocco

* ayoub.ezz1990@gmail.com

## Abstract

Tuberculosis (TB) remains a major public health challenge in Morocco. This study analyzed spatiotemporal patterns of TB cases in Tétouan, Morocco using routine surveillance data from 2019–2023 (3,614 cases). Temporal trends and seasonality were assessed using seasonal–trend decomposition, spatial dependence using Global Moran's I and Getis-Ord Gi*; and space–time clustering using Kulldorff's scan statistic. The average annual TB incidence rate was 113 per 100,000 population. Men accounted for 62.94% (2,272/3,614) of cases, with a mean age of 40.0 years (SD 19.1). Pulmonary TB (PTB) represented 56.71% (2,049/3,614) of TB cases, of which 89.87% (1,828/2,034) were bacteriologically confirmed. Notifications declined from 2019 to 2021, coinciding with COVID-19–related service disruptions, followed by a resurgence in 2022–2023. Seasonal peaks were observed mainly in spring and autumn. Spatial analyses identified a persistent high-incidence hotspot in the urban municipality of Tétouan. At the district level, space–time analyses using Monte Carlo permutation tests implemented in SaTScan confirmed significant PTB clustering in urban slum districts, with relative risks (RR) ranging from 1.90 to 2.03 (p < 0.01), suggesting stable reservoirs of TB transmission. Over time, clustering expanded beyond the urban core, with emerging TB hotspots in peri-urban communes (Azla, Zaitoune, and Beni Karrich). At the commune level, Azla and Zaitoune formed the most likely PTB cluster (RR 2.20 during 2022–2023; p < 0.001), while a secondary high-risk cluster was detected in a rural commune (RR up to 3.00; p < 0.01). These findings indicate that slums are entrenched reservoirs of transmission in urban settings, while peri-urban zones are emerging hotspots shaped by urbanization and mobility.

**Data availability statement:** All data required to replicate the findings of this study are provided within the manuscript and the Supporting Information files. S1 Text contains the detailed methodological protocol, case definitions, and analytical procedures. S1 Table summarizes the administrative subdivisions used during the study. Annual crude tuberculosis incidence rates by commune and urban district are presented in S2 Table, and annual crude pulmonary tuberculosis incidence rates are provided in S3 Table. S1 Data includes the de-identified individual-level tuberculosis case dataset along with district and commune coordinates. S2 Data contains the geographic shapefiles and SaTScan outputs used for spatial and spatio-temporal analyses.

**Funding:** The author(s) received no specific funding for this work.

**Competing interests:** The authors have declared that no competing interests exist.

Targeted strategies—intensified case detection and social support in slums, mobile diagnostics in peri-urban and rural areas, and seasonally aligned control— combined with routine spatiotemporal surveillance may help reduce geographic inequities and strengthen TB control efforts.

## Introduction

Tuberculosis (TB) remains a leading infectious cause of mortality worldwide [1]. In 2023, an estimated 10.8 million people developed TB and 1.25 million died, with the burden disproportionately affecting low- and middle-income countries [1].

Despite being preventable and curable, TB continues to be a disease of poverty driven by structural and social determinants [1]. Progress toward the World Health Organization (WHO) End TB Strategy targets has been slow and uneven, reflecting persistent gaps in transmission monitoring and in the implementation of effective interventions [2,3].

Urbanization is a key determinant of TB epidemiology [3]. Rapid city growth and rural-to-urban migration have expanded informal settlements and slums, where overcrowding, poor ventilation, limited hygiene, and restricted access to healthcare sustain *Mycobacterium tuberculosis* transmission [4]. Growing evidence shows that urban slums act as reservoirs of infection, with transmission extending to peri-urban and rural areas through commuting, seasonal migration, and social networks—a pattern increasingly documented in African and Asian cities [4–6].

In Morocco, TB remains a major public health concern [7]. Since the launch of the national TB program in the late 1970s, notable progress has been made, achieving a case detection rate of 83% and treatment success of 85.3% by 2015, in line with WHO targets [8]. However, TB incidence has declined only modestly, from 115 per 100,000 inhabitants in 2000–94 per 100,000 in 2021, with marked regional disparities [7]. In 2021, the Tangier–Tétouan–Al Hoceima region reported the highest incidence nationwide (111 per 100,000 inhabitants) [7]. Within this region, Tétouan exemplifies complex urban–rural interactions, encompassing a dense historic Old Medina with entrenched slum districts shaped by rural-to-urban migration, peri-urban communes undergoing rapid urbanizing demographic and spatial change, and rural villages increasingly connected to urban centers through mobility and commerce. While previous studies in Morocco have described broad TB epidemiological patterns, few have explicitly examined slums as transmission hubs or traced TB dynamics across the urban–rural continuum.

Pulmonary TB (PTB), particularly when bacteriologically confirmed, is the main driver of transmission, whereas extrapulmonary TB (EPTB) more often reflects reactivation or dissemination [9]. However, many spatiotemporal studies have analyzed TB cases without distinguishing between clinical forms, potentially obscuring areas of active transmission. Applying spatiotemporal analyses separately to PTB and EPTB can help identify high-risk transmission zones, characterize spatial diffusion patterns, and reveal seasonal or geographic heterogeneity linked to social, climatic,

and mobility-related determinants [10]. Such analyses can inform targeted interventions and guide resource allocation in heterogeneous settings.

This study aimed to analyze the spatial and temporal distribution of TB in Tétouan Province from 2019 to 2023, focusing on the role of urban slums in sustaining transmission and the subsequent spread of PTB into peri-urban and rural communes. By integrating fine-scale (district- and commune-level) multi-year spatiotemporal analyses with routine surveillance data, we sought to generate evidence to support more equitable and geographically targeted interventions under the national TB control program.

## Materials and methods

### Study setting

The study was conducted in Tétouan Province, located in the northwestern Mediterranean region of Morocco (35°34′42.42″N–5°22′6.13″W). Tétouan is one of the eight provinces of the Tangier–Tétouan–Al Hoceima region (S1 Fig). According to the 2019 administrative division High Commission for Planning (Haut Commissariat au Plan) [11], the province covers 2,541 km$^2$ and includes two urban municipalities (Tétouan and Oued Laou) and twenty rural communes with a total population of 573,784 inhabitants. The urban population accounts for 416,988 inhabitants, corresponding to an urbanization rate of 72.7%.

District-level analyses were conducted exclusively within the urban municipality of Tétouan, which is subdivided into 18 administrative districts. Seven districts form the Old Medina, characterized by high population density, overcrowded housing, poor ventilation, and limited infrastructure, and are commonly classified as slum-like areas. The remaining 11 districts constitute the New (modern) Medina. Detailed descriptions and names of districts and communes are provided in S1 Table.

### Data source

Data on all notified TB cases between 2019 and 2023 were obtained from the registers and clinical records of the Diagnostic Center of Tuberculosis and Respiratory Diseases of Tétouan (CDTMR), which operates under the National Tuberculosis Control Program (PNLAT). Data access granted on 13 March 2023.

For each patient, the registry captured sociodemographic information (age, sex, and place of residence), clinical form of TB (pulmonary or extrapulmonary), bacteriological status (GeneXpert *Ultra* MTB/RIF result), treatment history (new or previously treated), and treatment outcomes. Individual case records were extracted and aggregated by spatial unit of analysis (district for urban Tétouan or commune for province-wide analyses) and by time unit (month and year) to generate analytical datasets prior to statistical analysis using Microsoft Excel (S1 Data). Population denominators for incidence calculations were obtained from the Moroccan High Planning Commission (2019) [11] and Regional Directorates of Public Health.

### Case definitions

Tuberculosis cases were classified according to national and WHO guidelines. For the clinical form status: Pulmonary TB (PTB) included patients with clinical and radiological findings consistent with TB and was considered bacteriologically confirmed when GeneXpert *Ultra* MTB/RIF was positive (the primary diagnostic test in Tétouan since 2017). Smear microscopy and culture were used for bacteriological follow-up of patients. Extrapulmonary TB (EPTB) was defined as organ-specific disease diagnosed through histopathology, imaging, or bacteriological testing when available. Treatment history status was classified as new (no prior anti-tuberculosis treatment or treatment duration of <1 month) or previously treated (treatment duration ≥1 month), and further categorized as relapse, treatment after loss to follow-up, or treatment after failure. Treatment outcomes were recorded as cured, treatment completed, treatment failure, death, loss to follow-up,

transferred out/not evaluated, or diagnostic error (cases initially registered as TB and subsequently reclassified as non-TB) (S1 Text).

### Inclusion and exclusion criteria

All TB cases notified to the PNLAT registry in Tétouan between 2019 and 2023 were included (N = 3,614). No cases were excluded due to missing information. As missing data were limited and variable-specific, analyses were conducted using an available-case approach; denominators therefore vary across variables.

### Statistical analysis

**Time-series analysis.** Monthly TB incidence data were decomposed into seasonal, trend, and residual components using Seasonal-Trend decomposition based on LOESS (STL), implemented in R version 4.1.1 [12].

**Spatial autocorrelation.** Global Moran's I index was used to assess overall spatial clustering, while local Moran's I and Getis-Ord Gi* statistics were applied to identify spatial hot and cold spots. Spatial analyses and mapping were conducted using ArcGIS version 10.8 (ESRI, Redlands, CA, USA).

**Spatiotemporal scan analysis.** Kulldorff's retrospective space-time scan statistic [13] was implemented in SaTScan (version 10.2.5) using a discrete Poisson model. Monthly case counts were used as the temporal aggregation unit. The maximum spatial cluster size was set at 30% of the population at risk, and the maximum temporal cluster size at 50% of the study period. Statistical significance was assessed using Monte Carlo simulation with 999 replications. Clusters with the largest log-likelihood ratio (LLR) and $p < 0.05$ were considered most likely, while other significant clusters were classified as secondary. Relative risk (RR) for each cluster was calculated as the estimated risk within the cluster relative to the risk outside the cluster.

Full methodological details, including case definitions, missing data handling, geocoding procedures, formulas, spatial and spatiotemporal parameters, and cluster definitions, are provided in S1 Text.

### Ethical considerations

This study used anonymized routine surveillance data collected under the Moroccan National Tuberculosis Control Program. No personal identifiers were accessed or analyzed. The study protocol was approved by the Institutional Review Board of the Hospital–University Ethics Committee of Tangier (CEHUT) (IRB No: AC112JV/2–025), and the requirement for informed consent was waived. All procedures were conducted in accordance with the Declaration of Helsinki.

## Results

### Demographic and clinical characteristics

Between 2019 and 2023, a total of 3,614 TB cases were notified in Tétouan, corresponding to a mean annual notification rate (new and retreatment cases) of 122.1 per 100,000 population and a mean annual incidence (new cases) rate of 113 per 100,000 population. Men accounted for 62.94% (n = 2,272), yielding a male-to-female ratio of 1.70. The mean age was 40 years (SD 19.1). Most patients (82.63%, n = 2,936) resided in urban areas, of whom 37.0% (n = 1,085) were concentrated in Old Medina slum districts.

New cases represented 92.83% (n = 3,352) of the total. PTB accounted for 55.74% (n = 2,014), of which 89.87% (n = 1,828) were bacteriologically confirmed. EPTB accounted for 43.29% (n = 1,564), most commonly pleural (40.1%, n = 646) and lymph node TB (34.1%, n = 544).

Both clinical forms were concurrently diagnosed in 35 patients (0.97%).

Treatment outcomes were available for 3,505 patients (97.0%). The overall treatment success rate was 87.87%. Treatment failure occurred in 1.00% (n = 35), loss to follow-up in 6.45% (n = 226), and death in 2.94% (n = 103) (**Table 1**).

**Table 1. Demographic, clinical, and treatment characteristics of tuberculosis cases in Tétouan Province, 2019–2023 (N = 3,614).**

| Variables | n | % |
|---|---|---|
| **Sex** (n = 3610) | | |
| Male | 2272 | 62.94 |
| Female | 1338 | 37.06 |
| **Age group (years)** (n = 3612) | | |
| <18 | 318 | 8.80 |
| 18–30 | 1136 | 31.45 |
| 31–50 | 1094 | 30.29 |
| 51–65 | 615 | 17.03 |
| ≥65 | 449 | 12.43 |
| Mean age (SD), by years | 40 (19.1) | — |
| **Residence (address category)** (n = 3553) | | |
| Urban* | 2936 | 82.63 |
| *Old Medina (slum districts)* | *1085* | *37.00* |
| *New Medina* | *1851* | *63.00* |
| Rural | 571 | 16.07 |
| Homeless | 4 | 0.11 |
| Prison | 42 | 1.18 |
| **TB clinical form** (n = 3613) | | |
| PTB | 2014 | 55.74 |
| EPTB | 1564 | 43.29 |
| PTB+EPTB (concurrent) | 35 | 0.97 |
| **PTB confirmation status** (n = 2034) | | |
| Bacteriologically confirmed | 1828 | 89.87 |
| Clinically diagnosed | 206 | 10.13 |
| **EPTB localizations** (n = 1593) | | |
| Pleural | 646 | 40.10 |
| Lymph nodes | 544 | 34.15 |
| Peritoneal | 80 | 5.02 |
| Pericarditis | 49 | 3.08 |
| Osteo-articular | 47 | 2.95 |
| Mammary | 42 | 2.64 |
| Ascites | 34 | 2.13 |
| Cerebral | 27 | 1.69 |
| Uro-genital | 23 | 1.44 |
| Gastro-intestinal | 13 | 0.82 |
| Abscess | 8 | 0.50 |
| Other | 80 | 5.02 |
| **TB status** (n = 3611) | | |
| New cases | 3352 | 92.83 |
| Retreatment (total) ‡ | 259 | 7.17 |
| *After treatment failure* | *5* | *1.93* |
| *After lost to follow-up* | *77* | *29.73* |
| *Relapse* | *177* | *68.34* |
| **Treatment outcome** (n = 3505) | | |
| Treatment completed | 1957 | 55.83 |
| Cured | 1123 | 32.04 |

*(Continued)*

**Table 1.** (Continued)

| Variables | n | % |
|---|---|---|
| Failure | 35 | 1.00 |
| Diagnostic error | 15 | 0.43 |
| Lost to follow-up | 226 | 6.45 |
| Death | 103 | 2.94 |
| Transfer | 46 | 1.31 |

Data are presented as crude number and percentage unless otherwise indicated. *Percentages for Old/New Medina are calculated among urban cases (n = 2936), not among all residence categories. ‡Percentages for retreatment subcategories are calculated among retreatment cases (n = 259). Abbreviations: TB: Tuberculosis; PTB: Pulmonary Tuberculosis; EPTB: Extrapulmonary Tuberculosis; PTB+EPTB, concurrent pulmonary and extrapulmonary tuberculosis; SD, Standard Deviation.

### Trends and seasonal patterns of TB incidence

Overall TB incidence declined from 148.5 per 100,000 in 2019 to the lowest observed incidence of 109.3 per 100,000 in 2021, followed by an increase to 123.5 per 100,000 in 2023 (**Fig 1A**).

By clinical form, PTB incidence remained higher than EPTB throughout the study period. PTB incidence decreased from 81.2 per 100,000 in 2019 to 65.2 per 100,000 in 2021, before increasing to 70.0 per 100,000 in 2023. A comparable temporal pattern was observed for EPTB, with incidence declining from 67.3 per 100,000 in 2019 to 44.1 per 100,000 in 2021 and subsequently rising to 53.9 per 100,000 in 2023 (**Fig 1A**).

TB incidence was consistently higher in urban than in rural areas. In urban settings, incidence declined from 154.0 per 100,000 in 2019 to 131.2 per 100,000 in 2021, followed by a modest increase to 137.1 per 100,000 in 2023. In contrast, rural incidence decreased from 69.1 per 100,000 in 2019 to 52.6 per 100,000 in 2021, then increased substantially to 91.9 per 100,000 in 2023 (**Fig 1B**).

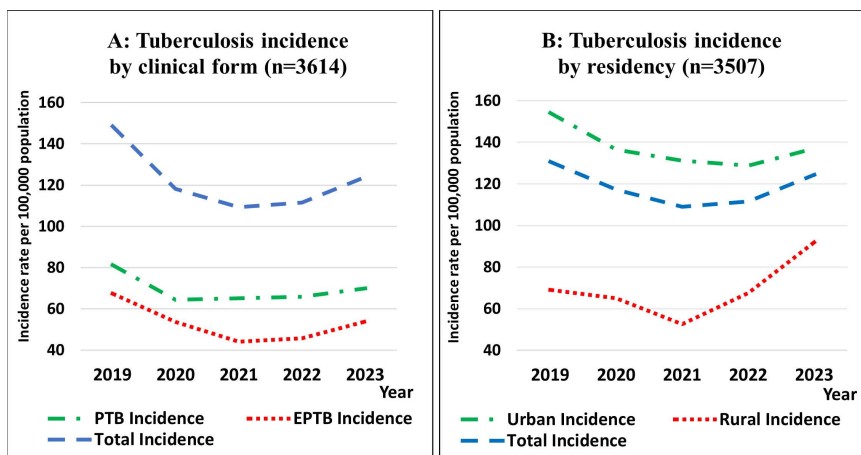

**Fig 1. Temporal trends in tuberculosis incidence by clinical form and residency from 2019 to 2023.** A: illustrates temporal trends in pulmonary tuberculosis (PTB), extrapulmonary tuberculosis (EPTB), and total TB incidence rates; B: shows incidence trends stratified by place of residence (urban vs. rural). Total incidence values differ between panels because they are calculated within each stratification and are not directly comparable.

Time-series decomposition of monthly TB incidence showed a decline from 2019 to 2021, followed by a gradual rebound through 2023. The seasonal component exhibited consistent peaks in March–May and secondary peaks in September–November, with lowest incidence in December (Fig 2).

## Spatial distribution of all TB and PTB cases

For all TB cases, commune-level mapping showed four communes with TB incidence >100 per 100,000: Tétouan Municipality (Mu) and the peri-urban rural counties of Beni Karrich, Azla, and Zaitoune. Tétouan Municipality had the highest incidence in 2019 (155 per 100,000), Beni Karrich peaked in 2020 (200 per 100,000), and Azla and Zaitoune increased from 2021 onward, reaching 193 and 215 per 100,000 in 2023, respectively (Fig 3A).

Global Moran's I indicated significant spatial clustering of TB incidence overall (Moran's $I_{2019–2023}$=0.706, p<0.001), with clustering detected in all years except 2020. Z-scores for all TB cases increased over the study period, indicating that spatial clustering became progressively stronger and more persistent over time (Table 2).

Within Tétouan Municipality, the highest district-level incidence occurred in Old Medina slum districts (Bab Tout, Mellah, Dersa III), while Touilaa had the highest incidence in the New Medina (S2 Table).

For PTB, Tétouan Municipality had the highest average annual incidence at the commune level (84.3 per 100,000), and among rural communes, Zaitoune (89.9 per 100,000) and Azla (57.9 per 100,000) were most affected (S3 Table).

The spatial distribution shifted over time: PTB incidence was highest in Tétouan Mu in 2019–2020, peaked in Beni Karrich in 2021 (117.2 per 100,000), and was highest in Azla and Zaitoune in 2022–2023 (Fig 3B). PTB showed no global clustering in 2019–2020, but significant positive spatial autocorrelation emerged from 2021 onward, with increasing Z-scores indicating progressively stronger clustering (Table 2).

District-level patterns mirrored all TB cases analysis, with the highest PTB incidence in Old Medina slum districts (Bab Tout, Mellah, and Dersa III) and the highest New Medina incidence in Touilaa district (S3 Table).

## Spatial cluster and hotspot analysis

Local spatial statistics showed consistent clustering for both all TB and PTB, with high–high areas concentrated in the northeastern part of the province (Fig 4). For all TB, high–high clusters included Tétouan Municipality and the adjacent communes of Azla, Zaitoune, and Beni Karrich, persisting across most years and strengthening in the cumulative 2019–2023 analysis (Fig 4A). PTB displayed a similar but delayed pattern: high–high clusters were largely restricted to Tétouan Municipality in 2019–2020, and then expanded from 2021 onward to include Azla and Zaitoune as persistent high–high areas (Fig 4B).

Low–low clusters for both outcomes were consistently located in the southwestern rural communes, particularly Bghaghza, Bni Idder, and El Kharroub, while Zinat appeared as a low–high outlier in the cumulative 2019–2023 analysis and Sahtryine (2020) and Saddina (2021) as year-specific outliers (Fig 4).

Getis–Ord Gi* hotspot analysis mirrored these patterns: hotspots for all TB were already present in 2019 around Tétouan Municipality and expanded to stable hotspots in Azla, Zaitoune, and Beni Karrich during 2021–2023 (Fig 5A), whereas PTB hotspots remained urban-focused until 2021 and became established in Azla and Zaitoune in 2022–2023 (Fig 5B). Cold spots decreased over time in both analyses and became increasingly concentrated in the southwest, mainly in Bghaghza and Bni Idder (Fig 5).

**Space–Time clusters of tuberculosis.** SaTScan identified significant space–time clustering for both all TB and PTB (Tables 3, 4).

At the district level, clusters were concentrated in Old Medina slum districts. For all TB cases, the most likely cluster included El Kassaba, Bab Tout, Mellah, Sidi Frij, and Touilaa (RR=1.78, p<0.001), with secondary clusters in El Kassaba–Bab Tout–Mellah–Sidi Frij (RR=1.68, p<0.001) and Dersa III (RR=1.68, p=0.012) (Table 3). For PTB, the most likely district-level cluster included El Kassaba, Bab Tout, Mellah, and Sidi Frij (RR=2.03; p<0.001). Two secondary

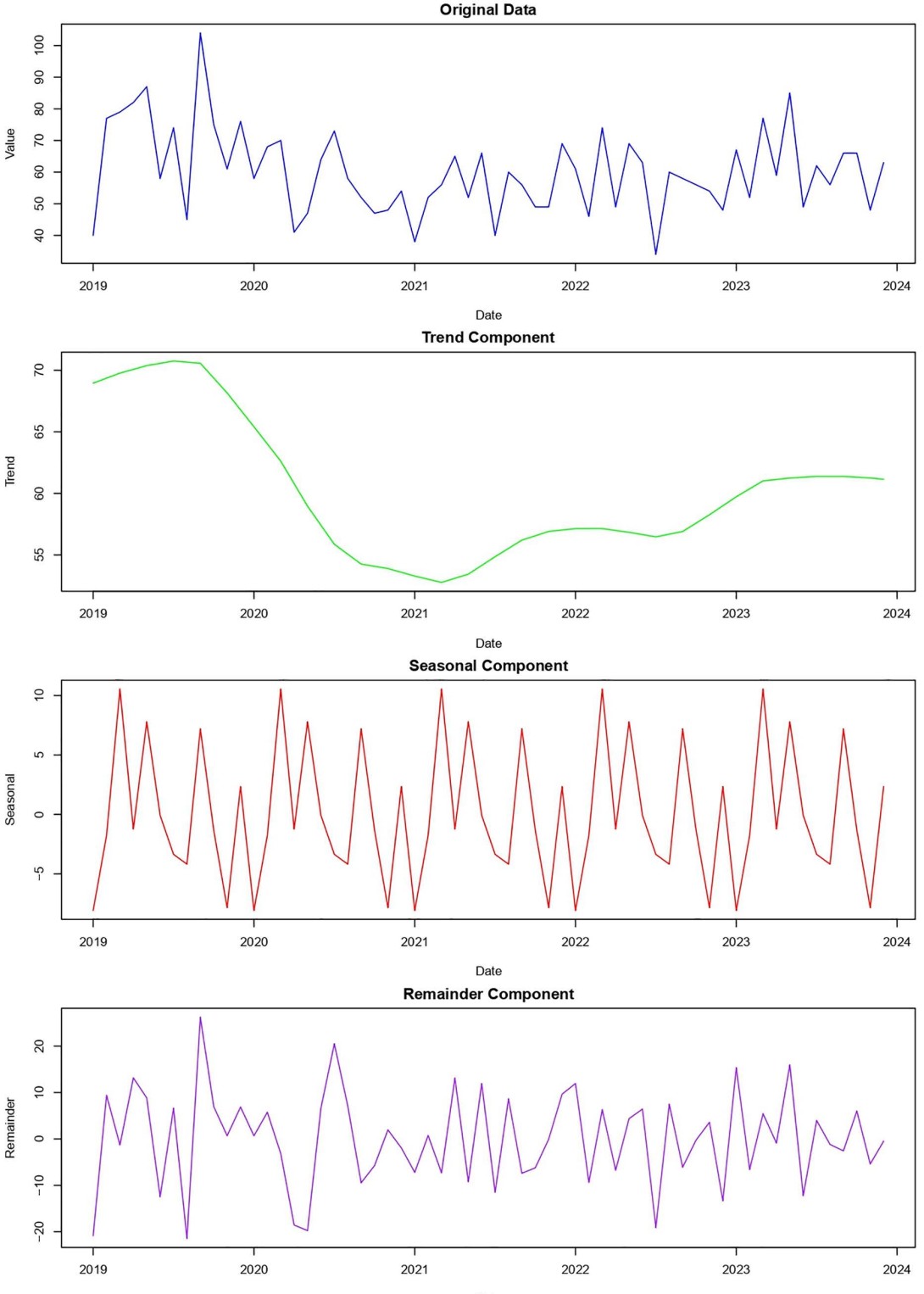

**Fig 2. Time series decomposition of reported tuberculosis cases in Tétouan, 2019–2023.** The top panel shows the original observed data over time. The second panel represents the trend component, illustrating the long-term underlying pattern after smoothing short-term fluctuations. The third panel displays the seasonal component, capturing recurrent periodic variations across the observation period. The bottom panel shows the remainder (residual) component, representing irregular fluctuations not explained by the trend or seasonal patterns.

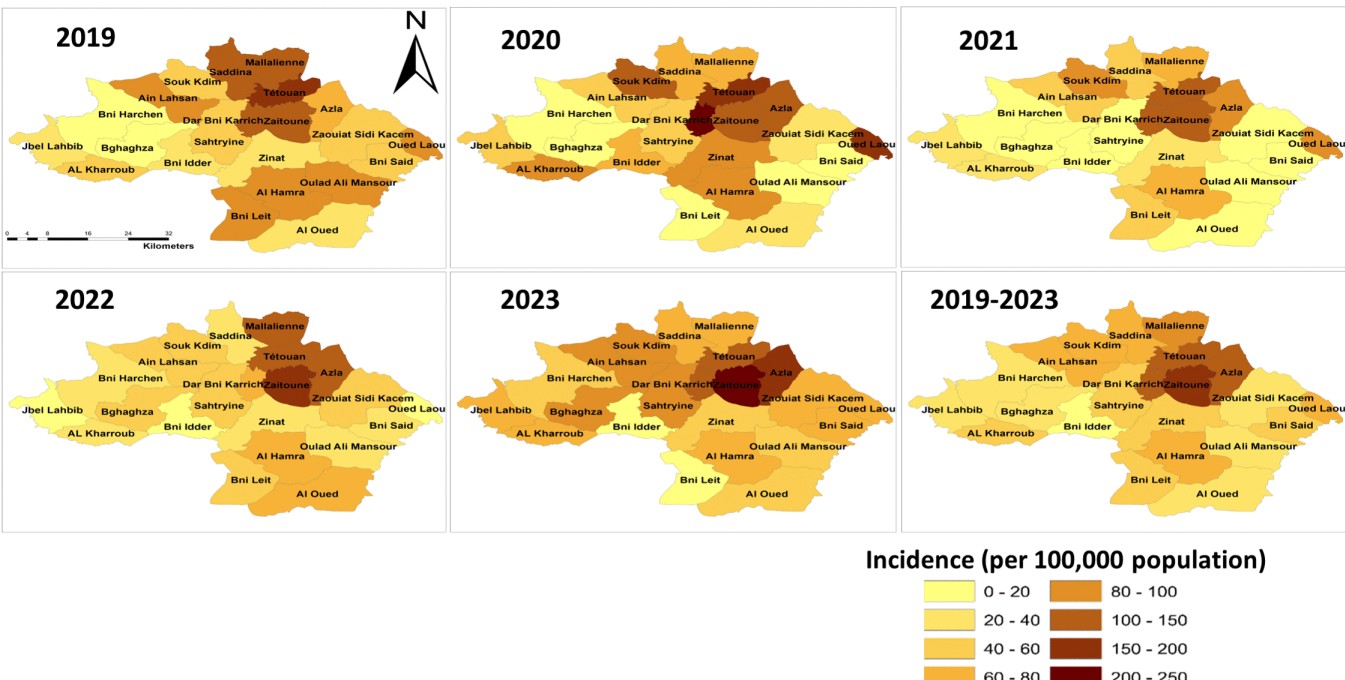

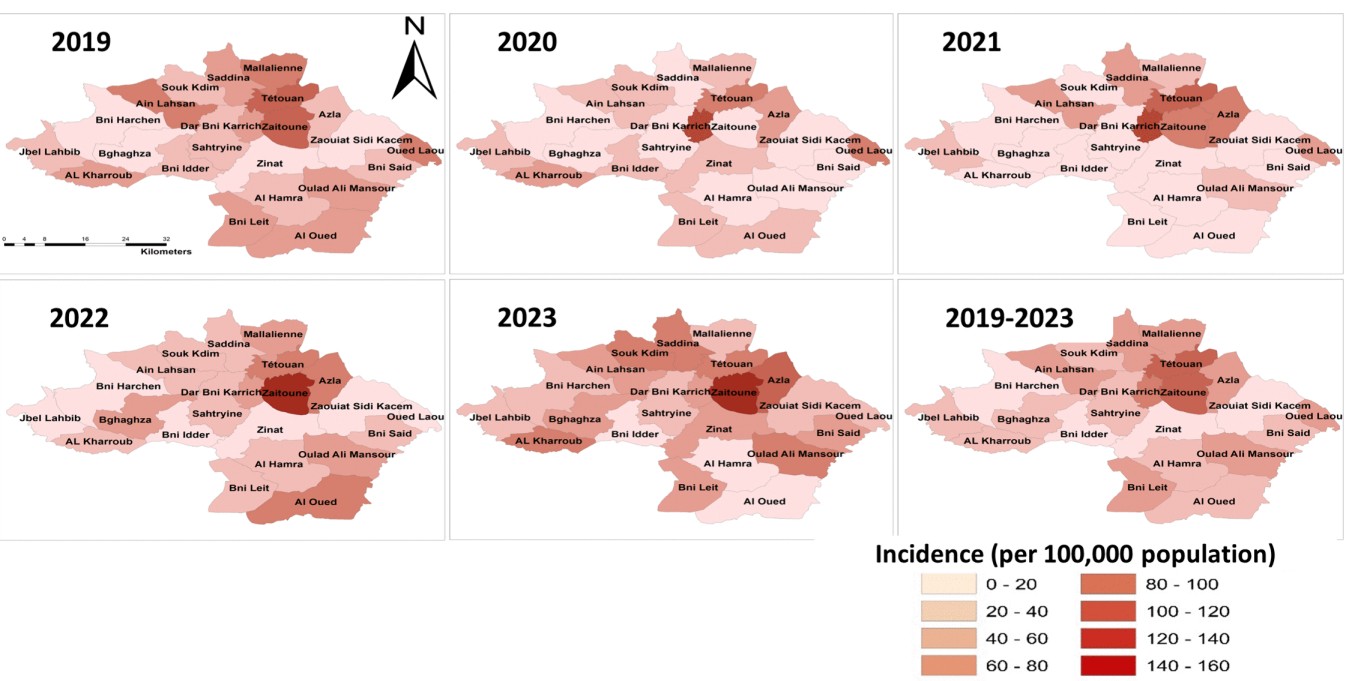

**Fig 3. Spatial distribution of tuberculosis incidence in Tétouan Province, Morocco, 2019–2023.** Incidence rate maps of tuberculosis across communes of Tétouan Province during the study period. **(A)** Incidence rates of all tuberculosis cases per 100,000 population; **(B)** Incidence rates of pulmonary tuberculosis per 100,000 population. Maps display annual incidence for each commune from 2019 to 2023, as well as the mean annual incidence for the entire study period (2019–2023). Incidence rates were calculated using the number of notified tuberculosis cases and corresponding population

estimates for each commune. Administrative boundaries derived from publicly available Morocco administrative boundary datasets distributed via the Humanitarian Data Exchange (United Nations Office for the Coordination of Humanitarian Affairs), https://data.humdata.org/dataset/cod-ab-mar, licensed under CC BY 4.0. These datasets compile official administrative boundary information originating from Moroccan national statistical sources, including the Haut-Commissariat au Plan. Commune boundaries were adapted from publicly available administrative boundary data. Maps were generated using ArcGIS (Esri).

**Table 2. Global spatial autocorrelation analysis for annual TB notification rate in Tétouan, Morocco from 2019 to 2023.**

| Year | Moran's I | Z-score | p-value | Pattern |
|---|---|---|---|---|
| **All Tuberculosis cases** | | | | |
| 2019 | 0.41 | 2.43 | 0.015 | Clustered |
| 2020 | 0.27 | 1.87 | 0.062 | Not clustered |
| 2021 | 0.42 | 2.71 | 0.007 | Clustered |
| 2022 | 0.54 | 3.53 | <0.001 | Clustered |
| 2023 | 0.54 | 3.59 | <0.001 | Clustered |
| 2019–2023* | 0.71 | 4.37 | <0.001 | Clustered |
| **Pulmonary Tuberculosis cases** | | | | |
| 2019 | 0.19 | 1.35 | 0.140 | Not clustered |
| 2020 | −0.13 | −0.47 | 0.641 | Not clustered |
| 2021 | 0.40 | 2.72 | 0.007 | Clustered |
| 2022 | 0.33 | 2.27 | 0.023 | Clustered |
| 2023 | 0.32 | 2.21 | 0.027 | Clustered |
| 2019–2023* | 0.55 | 3.47 | <0.001 | Clustered |

*The "2019–2023" row represents the pooled (cumulative) analysis using notification rates calculated over the full 5-year period (i.e., combining all years to assess the overall spatial dependence, not an additional single calendar year).

clusters were identified: one encompassing these districts plus Touilaa (RR = 1.78; $p < 0.001$), and another restricted to Dersa III (RR = 1.90; $p = 0.008$) (**Table 4**).

At the commune level, all TB clustering was strongest in Azla and Zaitoune (RR = 1.62, p = 0.003), with a secondary cluster in Oulad Ali Mansour (RR = 1.69, $p = 0.017$) (**Table 3**). For PTB, Azla and Zaitoune formed the most likely cluster (RR = 2.20, $p < 0.001$), and Oulad Ali Mansour showed two significant secondary clusters (RR = 2.84, $p = 0.012$; RR = 3.00, $p = 0.006$) (**Table 4**).

## Discussion

This study provides the first fine-scale (district/commune), multi-year spatiotemporal analysis of TB in northern Morocco, focusing on Tétouan Province. By combining all-TB and PTB-specific analyses, it highlights a persistent urban slum reservoir and an outward expansion of higher PTB risk toward peri-urban and rural communes.

The burden was concentrated among young adult men, consistent with national [14] and global [15] trends. Such predominance in men may be explained by occupational exposures, higher prevalence of smoking and substance use, delays in care-seeking behavior, and possible biological susceptibility (including immune/hormonal differences) [3,16,17].

TB notifications declined between 2019 and 2021 and rebounded in 2022–2023, reflecting COVID-19–related disruptions to TB control locally [16] and globally [18]. The subsequent increase likely reflects recovery of case detection and catch-up diagnosis of previously missed cases [18]. PTB accounted for 56.70% of cases, consistent with national estimates [14]. Among PTB cases, 89.87% were bacteriologically confirmed, suggesting substantial transmission potential

## A. Local spatial clusters of all tuberculosis incidence

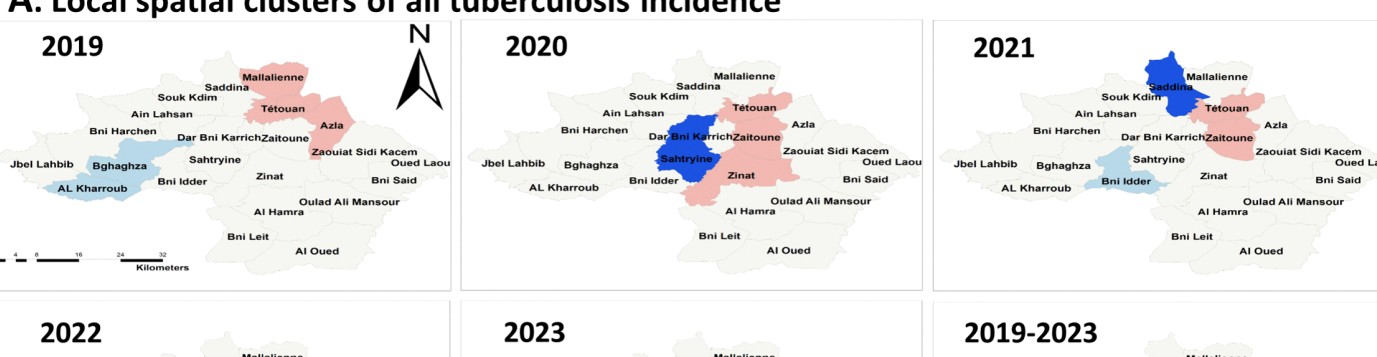
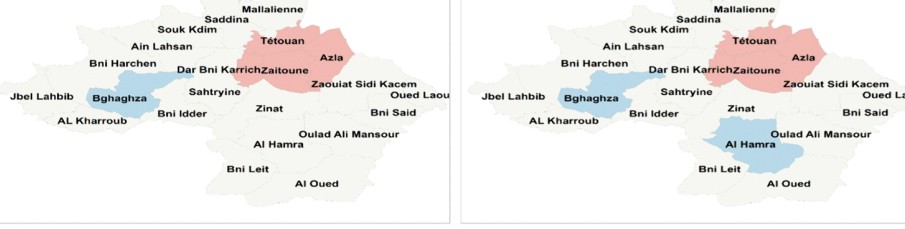
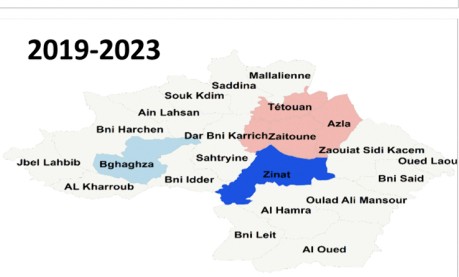

## B. Local spatial clusters of pulmonary tuberculosis incidence

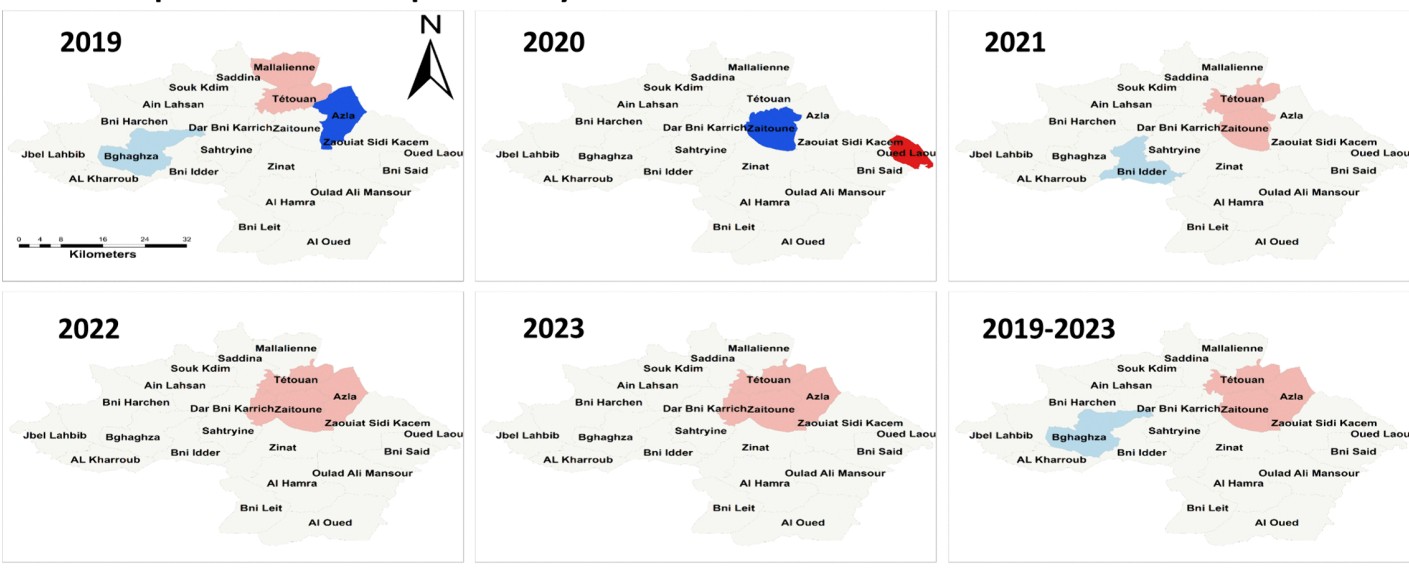
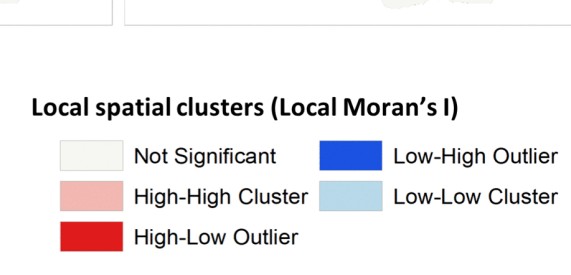

**Local spatial clusters (Local Moran's I)**

Not Significant — Low-High Outlier
High-High Cluster — Low-Low Cluster
High-Low Outlier

**Fig 4. Spatiotemporal patterns of tuberculosis incidence clusters in Tétouan Province, Morocco, 2019–2023.** Local spatial clusters of tuberculosis incidence identified using Local Moran's I analysis. **(A)** Spatial clusters of all tuberculosis incidence; **(B)** Spatial clusters of pulmonary tuberculosis incidence. Maps show yearly cluster patterns from 2019 to 2023, as well as clusters based on the mean incidence for the entire study period

(2019–2023). High–High clusters indicate communes with high incidence surrounded by neighboring communes with similarly high incidence, while Low–Low clusters indicate areas of consistently low incidence. High–Low and Low–High areas represent spatial outliers where local incidence differs from surrounding communes. Non-significant areas did not show statistically significant spatial autocorrelation. Administrative boundaries derived from publicly available Morocco administrative boundary datasets distributed via the Humanitarian Data Exchange (United Nations Office for the Coordination of Humanitarian Affairs), https://data.humdata.org/dataset/cod-ab-mar, licensed under CC BY 4.0. These datasets compile official administrative boundary information originating from Moroccan national statistical sources, including the Haut-Commissariat au Plan. Commune boundaries were adapted from publicly available administrative boundary data. Maps were generated using ArcGIS (Esri).

[19] and robust diagnostic ascertainment, supported by GeneXpert-based testing introduction since 2017. PTB incidence declined in 2019–2020, likely resulting from reduced transmission combined with under-detection during pandemic-related restrictions and diagnostic delays [16,18], followed by an increase as diagnostic services recovered. In contrast, EPTB declined through 2021 and increased modestly from 2022, plausibly reflecting its more complex diagnostic pathways (imaging/biopsy and specialist referral) compared with PTB [19–21]. Pleural and lymph node TB were the most frequent EPTB forms, likely reflecting dissemination from pulmonary sources [22].

Treatment success reached 87.87%, meeting the WHO End TB 2022 milestone target [23]. Yet loss-to-follow-up (6.45%) stands out as the critical weakness, fueling ongoing transmission and drug resistance—a challenge likely intensified by COVID-19 disruptions [16,24]— and also may reflect long-standing socioeconomic and healthcare access barriers [16,25]. Mortality was low (2.94%), below the national average [7] but still above WHO expectations [23], highlighting the need to strengthen retention strategies and treatment adherence.

Time-series decomposition revealed seasonal peaks in March–May and September–November, with declines during summer. Similar cycles have been observed in Algeria [26], Spain [27], Pakistan [28], and Iran [29], though they contrast with summer peaks in Kuwait [30] and India [31], underscoring the region-specific nature of TB seasonality. In northern Morocco, these seasonal fluctuations may be driven by climatic factors, including higher humidity and lower temperatures [32]. Additionally, reduced healthcare-seeking behavior during school examination periods and holidays may further contribute to the observed patterns [27,33].

Tétouan's urban area consistently ranked among the top three most affected counties during the study period. This pattern is likely driven by high population density (7,913 inhabitants/km$^2$ [11]), intense social mobility, and most importantly, the existence of overcrowded slums characterized by poor ventilation, limited sunlight, and inadequate sanitation [3]. Spatial clustering analyses of pulmonary TB and all TB cases identified several clusters predominantly encompassing slum areas within the historic medina (all TB: RR 1.68–1.78; PTB: RR 1.90–2.03), confirming a persistent concentration of cases in these neighborhoods. These findings align with the established role of slums in facilitating TB transmission globally and especially in African contexts [4–6]. Patterns of population movement—including rural-to-urban migration and cross-border flows from high TB-burden African countries—may further exacerbate exposure and transmission risk within the city's slums [3,4,6,34].

At the commune level, aggregated TB hotspots were identified in the northeastern region of Tétouan Province, encompassing Tétouan urban municipality and the adjacent rural communes of Azla, Zaitoune, and Beni Karrich, with high Getis-Ord Gi* hotspot confidence (95–99%). SaTScan revealed a primary cluster in Azla and Zaitoune (RR = 1.62, p = 0.003), likely reflecting rapid peri-urbanization and increased rural–urban mobility [4,34,35]. Similar urban clustering has been reported in Malaysia [36] and Bangladesh [5], although studies in Kenitra, Morocco [37] and China [38] suggest that rural predominance can also occur. For PTB, spatiotemporal patterns largely mirrored those of all TB cases but with higher relative risks (RR up to 2.20). PTB clusters frequently preceded or overlapped with clusters of all TB cases combined and, from 2021 onward, expanded beyond the urban core into peri-urban rural areas. This pattern highlights the central role of PTB in sustaining transmission and seeding new hotspots, particularly given that 89.87% of PTB cases were bacilliferous.

## A. Spatial hotspots and cold spots of all tuberculosis incidence

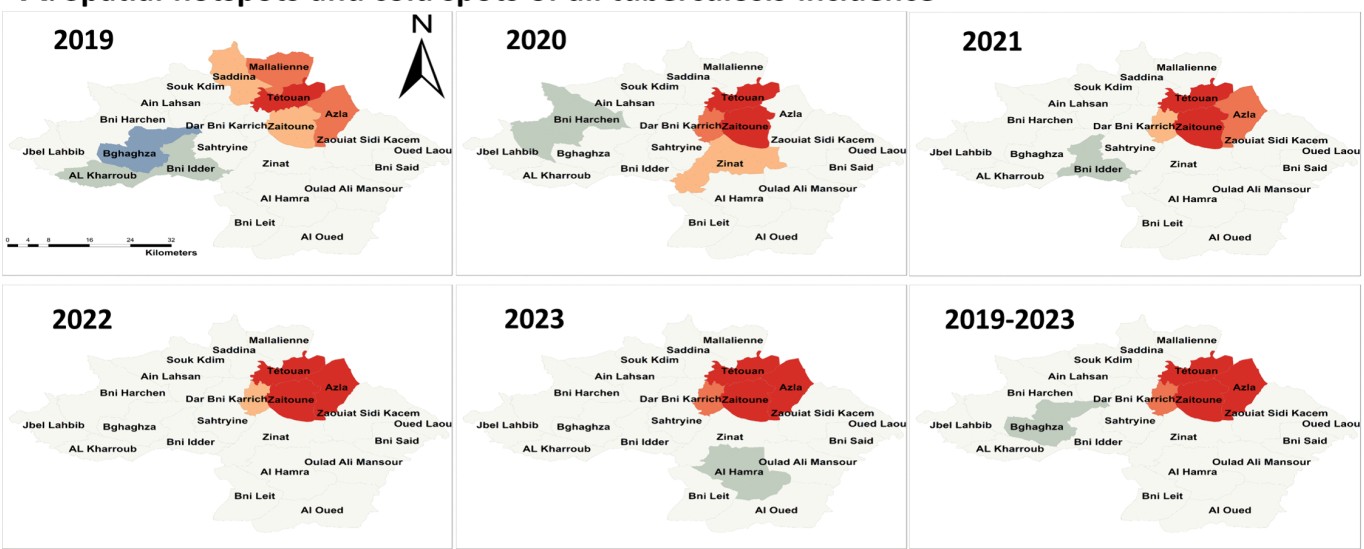

## B. Spatial hotspots and cold spots of pulmonary tuberculosis incidence

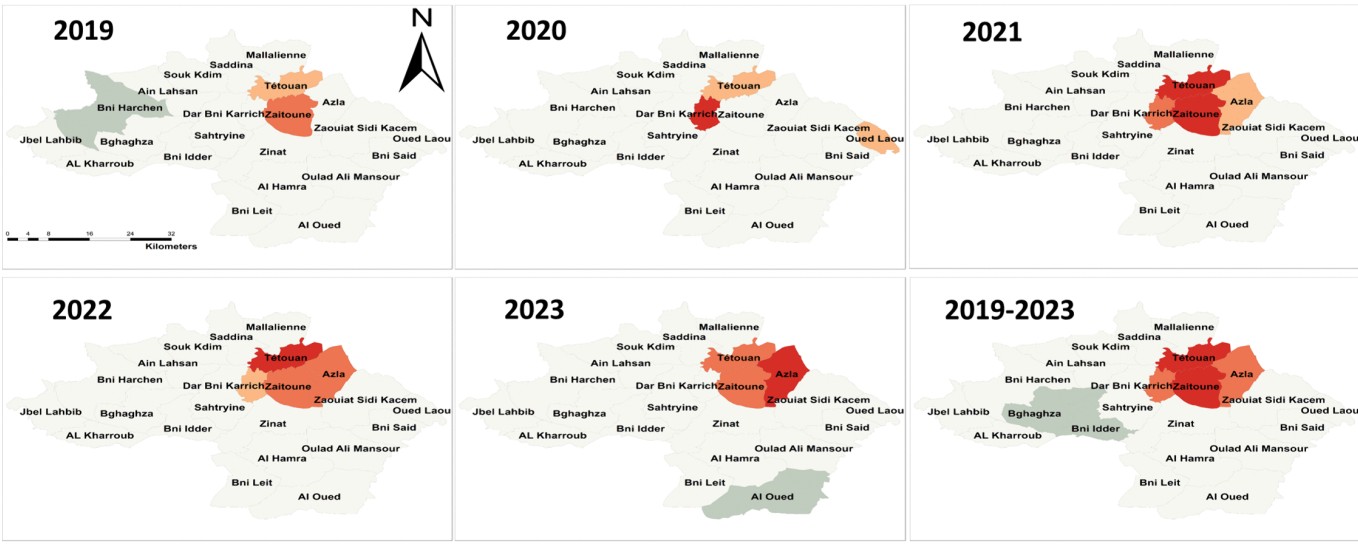

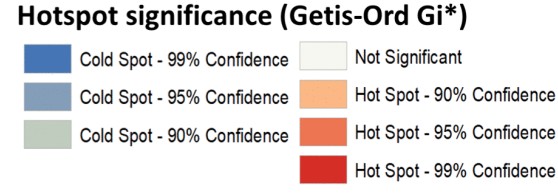

**Hotspot significance (Getis-Ord Gi\*)**

- Cold Spot - 99% Confidence
- Cold Spot - 95% Confidence
- Cold Spot - 90% Confidence
- Not Significant
- Hot Spot - 90% Confidence
- Hot Spot - 95% Confidence
- Hot Spot - 99% Confidence

**Fig 5. Spatial hotspots and cold spots of tuberculosis incidence in Tétouan Province, Morocco, 2019–2023.** Spatial hotspot analysis of tuberculosis incidence using the Getis-Ord Gi\* statistic. **(A)** Hotspot and cold spot clusters of all tuberculosis incidence. **(B)** Hotspot and cold spot clusters of pulmonary tuberculosis incidence. Maps show yearly hotspot patterns from 2019 to 2023, as well as clusters based on the mean incidence for the entire

study period (2019–2023). Red shades represent statistically significant hotspots (areas with high incidence surrounded by high-incidence neighbors), while blue shades represent significant cold spots (areas with low incidence surrounded by low-incidence neighbors). Confidence levels correspond to 90%, 95%, and 99% significance thresholds, while grey areas indicate locations with no statistically significant clustering. Administrative boundaries derived from publicly available Morocco administrative boundary datasets distributed via the Humanitarian Data Exchange (United Nations Office for the Coordination of Humanitarian Affairs), https://data.humdata.org/dataset/cod-ab-mar, licensed under CC BY 4.0. These datasets compile official administrative boundary information originating from Moroccan national statistical sources, including the Haut-Commissariat au Plan. Commune boundaries were adapted from publicly available administrative boundary data. Maps were generated using ArcGIS (Esri).

**Table 3. Space–time clusters of all tuberculosis cases in Tétouan, Morocco, 2019–2023 identified by SaTScan analysis.**

| Cluster Type | Number of Clustering areas | Cluster districts or communes | Time frame (Month/Year) | Observed cases | Expected cases | Relative risk (RR) | Log likelihood ratio (LLR) | P value |
|---|---|---|---|---|---|---|---|---|
| At the District level in Urban area (Tétouan commune) | | | | | | | | |
| Most likely cluster | 5 | El Kassaba, Bab Tout, Mellah, Sidi Frij, Touilaa | 02/2019 to 08/2020 | 252 | 147.31 | 1.78 | 32.620354 | < 0.001 |
| Secondary cluster 1 | 4 | El Kassaba, Bab Tout, Mellah, Sidi Frij | 12/2022 to 03/2023 | 238 | 146.57 | 1.68 | 25.479151 | < 0.001 |
| Secondary cluster 2 | 1 | Dersa III | 04/2019 to 06/2021 | 103 | 62.33 | 1.68 | 11.359865 | 0.012 |
| At the communes level (Tétouan Province) | | | | | | | | |
| Most likely cluster | 2 | Azla, Zaitoune | 07/2022 to 12/2023 | 87 | 54.31 | 1.62 | 8.46 | **0.003** |
| Secondary cluster | 1 | Oulad Ali Mansour | 01/2019 to 12/2023 | 62 | 36.88 | 1.69 | 7.18 | **0.017** |

Relative risk (RR) represents the estimated risk of tuberculosis within the cluster relative to the risk outside the cluster. The log likelihood ratio (LLR) indicates the strength of the cluster, with higher values reflecting stronger evidence of clustering. P-values were calculated using 999 Monte Carlo replications in SaTScan. Abbreviations: RR = relative risk; LLR = log likelihood ratio.

Notably, a secondary TB cluster was identified in Oulad Ali Mansour County, with a relative risk of 1.69 for all TB cases and up to 3.00 for PTB. Despite the commune's low population density (48.84 inhabitants/km$^2$ [11]), its mountainous terrain, limited access to the TB diagnostic center, dispersed yet overcrowded households, and reliance on a single health facility create conditions that may facilitate TB persistence and spread once introduced [39]. Alternatively, the cluster may reflect localized transmission amplification (e.g., delayed diagnosis, congregate exposure, or a small number of highly connected infectious cases). Further molecular and genomic studies are needed to confirm these hypotheses and guide interventions [40].

Conversely, low–low clusters and cold spots were predominantly detected in southwestern areas, particularly in Bghaghza County. The area's relatively low population density (71.32 inhabitants/km$^2$ [11]), abundant green spaces, mountainous terrain, weaker connectivity to urban hotspots, and better local access to healthcare services likely contribute to its lower TB incidence [41,42]. The persistence of cold spots in this region is consistent with an urban-centered TB burden and limited transmission into more remote rural communes.

Taken together, our findings delineate three epidemiological zones requiring differentiated strategies. First, urban slums represent critical transmission nodes and require continuous case finding, strengthened and systematic contact tracing, community education on housing conditions (ventilation, crowding), hygiene, and the benefits of smoking cessation, as well as targeted social support (e.g., transport or food assistance) to reduce diagnostic delays and improve treatment retention. Second, peri-urban communes (e.g., Azla, Zaitoune, and Beni Karrich) are emerging hotspots and should be prioritized for intensified surveillance, mobile diagnostic units — including seasonal outreach days offering on-site sputum GeneXpert testing and chest X-rays — strengthened patient referral and transport networks to diagnostic centers, and

**Table 4. Pulmonary tuberculosis spatial-temporal clusters detected using SaTScan in Tétouan, Morocco from 2019 to 2023.**

| Cluster Type | Number of Cluster-ing areas | Cluster districts or communes | Time frame (Month/Year) | Observed cases | Expected cases | Relative risk (RR) | Log likelihood ratio (LLR) | P value |
|---|---|---|---|---|---|---|---|---|
| At the District level in Urban area (Tétouan commune) | | | | | | | | |
| Most likely cluster | 4 | El Kassaba, Bab tout, Mel-lah, Sidi Frij | 2020/12–2023/5 | 170 | 88.51 | 2.03 | 31.651110 | < 0.001 |
| Secondary cluster 1 | 5 | El Kassaba, Bab tout, Mel-lah, Sidi Frij, Touilaa | 2019/2/1–2020/8/31 | 142 | 82.93 | 1.78 | 18.439717 | < 0.001 |
| Secondary cluster 2 | 1 | Dersa III | 2019/12/1–2022/5/31 | 73 | 39.33 | 1.90 | 11.839252 | 0.008 |
| At the communes level (Tétouan Province) | | | | | | | | |
| Most likely cluster | 2 | Azla, Zaitoune | 2022/1/1–2023/4/30 | 59 | 27.31 | 2.20 | 14.019637 | < 0.001 |
| Secondary cluster 1 | 1 | Oulad Ali Mansour | 2022/1/1–2023/4/30 | 22 | 7.39 | 3.00 | 9.451602 | 0.006 |
| Secondary cluster 2 | 1 | Oulad Ali Mansour | 2019/1/1–2021/4/30 | 23 | 8.17 | 2.84 | 9.028883 | 0.012 |
| Secondary cluster 3 | 1 | Beni Karrich | 2020/1/1–2021/4/30 | 20 | 9.41 | 2.14 | 4.514268 | 0.408 |

Relative risk (RR) represents the estimated risk of tuberculosis within the cluster relative to the risk outside the cluster. The log likelihood ratio (LLR) indicates the strength of the cluster, with higher values reflecting stronger evidence of clustering. P-values were calculated using 999 Monte Carlo repli-cations in SaTScan. Abbreviations: RR = relative risk; LLR = log likelihood ratio.

management of latent TB infection [43]. Recurrent clustering in Oulad Ali Mansour County indicates an elevated outbreak risk; integrating molecular epidemiology with routine spatiotemporal monitoring could support rapid, targeted responses [38,39]. Third, low-risk southwestern communes warrant ongoing monitoring, community awareness, and robust referral pathways to preserve low incidence and enable rapid investigation of any PTB increase to prevent sudden outbreaks from imported strains [3].

Integrating routine surveillance with spatiotemporal analysis—and, where feasible, molecular genotyping—can strengthen TB program performance by guiding resource allocation, enabling early active case-finding during sea-sonal peaks, and anticipating emerging hotspots. Given that PTB drives transmission, monitoring the expansion of PTB hotspots can inform targeted interventions in communes that function as transmission "bridges" between urban and rural areas. In parallel, differentiated adherence support—including community follow-up, tracing of treatment interruptions, and patient-centred measures such as transport support and appointment reminders—may reduce loss to follow-up in both high-burden urban settings and geographically remote communes.

Although this study focused on Tétouan, the results are likely transferable to other rapidly urbanizing cities where slum growth, migration, and limited health services sustain TB transmission [4–6,35,36,44].

Several limitations should be acknowledged. COVID-19 disruptions may have caused underreporting, affecting incidence estimates and clustering results. The ecological nature of the spatial analyses precludes inference on individual-level trans-mission. Some socioeconomic and clinical data (e.g., income, education, comorbidities) were unavailable. The absence of molecular data prevented validation of whether clusters represented true transmission chains, limiting causal inference.

In conclusion, TB in Tétouan exhibits spatial clustering and seasonality, driven by social and environmental factors. PTB remains the main driver of transmission, with hotspots expanding from urban slums to peri-urban rural areas. Sus-tained spatiotemporal surveillance, combined with geographically and temporally targeted interventions, will be critical for efficient TB control and may guide strategies in similar settings globally. Prioritizing high-burden slum and emerging peri-urban areas can also help reduce inequities in access to timely TB care.

**Supporting information**

**S1 Fig. Geographic location and administrative organization of Tétouan, Morocco.** A: Map of Morocco with regional organization; B: Region of Tangier-Tétouan-Al Hoceima; C: Province of Tétouan divided into communes.
(TIF)

**S1 Table. Administrative Division of Tétouan Province, by the High Commission for Planning, Tangier-Tétouan-Al Hoceima Regional Directorate, 2018.** This table summarizes the administrative organization of Tétouan Province. Urban areas include Tétouan and Oued Laou municipalities. Tétouan is subdivided into the Old Medina (historic, high-density neighborhoods including slums) and the New Medina (modernized areas with improved housing). Oued Laou, though urban, has a small population (11,690 in 2014), a limited area (~32.8 km$^2$), no district subdivision, and one primary health-care center. Rural areas consist of 20 surrounding communes.
(DOCX)

**S2 Table. Annual crude tuberculosis incidence rates (per 100,000 population) by commune and urban district in Tétouan Province, Morocco, 2019–2023.** The table presents annual crude tuberculosis incidence rates in Tétouan Province. Urban areas include Tétouan and Oued Laou municipalities. Tétouan municipality is divided into the Old Medina (historic, high-density neighborhoods, all located within the ancient walled Medina) and the New Medina (modern areas with more recent urban planning). Oued Laou, though urban, has a small population, limited area (~32.8 km$^2$), no district subdivision, and one primary healthcare center. Rural areas comprise 20 surrounding communes. Rates are expressed per 100,000 population; zero indicates no reported cases. The 2019–2023 values represent the average annual incidence over the five-year period, calculated using reported cases and population estimates from the Moroccan High Planning Commission (HCP) and regional health records.
(DOCX)

**S3 Table. Annual crude pulmonary tuberculosis incidence rates (per 100,000 population) by commune and urban district in Tétouan Province, Morocco, 2019–2023. Legend:** This table presents annual crude pulmonary tuberculosis incidence rates in Tétouan Province. Urban areas include Tétouan and Oued Laou municipalities. Tétouan is divided into the Old Medina (historic, high-density neighborhoods, all located within the ancient walled Medina) and the New Medina (modern areas with more recent urban planning). Oued Laou, though urban, has a small population, limited area (~32.8 km$^2$), no district subdivision, and one primary healthcare center. Rural areas comprise 20 surrounding communes. Rates are expressed per 100,000 population; zero indicates no reported cases. The 2019–2023 values represent the average annual incidence over the five-year period, calculated using reported cases and population estimates from the Moroccan High Planning Commission (HCP) and regional health records.
(DOCX)

**S1 Data. Individual-level tuberculosis case data, Tétouan Province, Morocco, 2019–2023.** Legend: The dataset contains individual-level information for all tuberculosis cases notified in Tétouan Province, Morocco, from January 2019 to December 2023. All variables included in the excel file: year and month of notification, date of registration, gender, TB case status (new case/retreatment), age, residential address(urban/rural and districts for urban residency), pulmonary tuberculosis (PTB) classification, extrapulmonary TB (EPTB) localization, sputum smear microscopy results, treatment outcome, treatment regimen. Data were collected from the Provincial Diagnostic Center for TB in Tétouan and represent routine surveillance data from the National Tuberculosis Control Program.
(XLSX)

**S2 Data. SaTScan output and Tétouan boundary shapefiles used for spatial tuberculosis analyses.** Legend: This compressed folder contains all spatial analysis inputs and outputs used in the study. It includes: (i) SaTScan result files

for space–time cluster detection of notified tuberculosis cases in Tétouan Province and (ii) the ArcGIS-compatible polygon shapefile of Tétouan and its administrative units used to map case locations and clusters. These files correspond to the spatial and spatiotemporal analyses reported in the manuscript and allow full reproduction of the cluster detection and mapping procedures.
(RAR)

**S1 Text. Extended materials, methods, and operational definitions for tuberculosis surveillance and spatial analyses in Tétouan Province, 2019–2023.** Legend: This file provides additional details on case definitions, inclusion criteria, handling of missing data, and analytical methods (time-series, spatial, and spatio-temporal analyses) used in the study.
(DOCX)

## Author contributions

**Conceptualization:** Nadya Mezzoug, Abdnour Boulaich, Khalid Bouti, Noureddine Elmtili.

**Data curation:** Ayoub Ez-Zari, Zaida Herrador, Laila Farouk, Abdnour Boulaich, Khalid Bouti, Zakaria Mennane, Noureddine Elmtili.

**Formal analysis:** Ayoub Ez-Zari, Laila Farouk, Nadya Mezzoug.

**Investigation:** Ayoub Ez-Zari, Zakaria Mennane, Noureddine Elmtili.

**Methodology:** Ayoub Ez-Zari, Zaida Herrador, Abdnour Boulaich, Khalid Bouti, Zakaria Mennane, Noureddine Elmtili.

**Project administration:** Nadya Mezzoug, Noureddine Elmtili.

**Resources:** Laila Farouk.

**Software:** Ayoub Ez-Zari, Zaida Herrador, Laila Farouk, Khalid Bouti.

**Supervision:** Ayoub Ez-Zari, Zaida Herrador, Abdnour Boulaich, Khalid Bouti, Zakaria Mennane, Noureddine Elmtili.

**Validation:** Ayoub Ez-Zari, Zaida Herrador, Nadya Mezzoug, Abdnour Boulaich, Khalid Bouti, Zakaria Mennane, Noureddine Elmtili.

**Visualization:** Ayoub Ez-Zari, Zaida Herrador, Nadya Mezzoug, Khalid Bouti.

**Writing – original draft:** Ayoub Ez-Zari, Laila Farouk.

**Writing – review & editing:** Zaida Herrador, Nadya Mezzoug, Abdnour Boulaich, Zakaria Mennane, Noureddine Elmtili.

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
