## [Decision Letter · Decision Letter 0]

18 Dec 2025

PGPH-D-25-02866

Spatiotemporal patterns of tuberculosis within the urban slums and expanding urban–rural transition zones: lessons from Tétouan, Morocco (2019–2023)

Dear Dr. Ez-Zari,

Thank you for submitting your manuscript to PLOS Global Public Health. After careful consideration, we feel that it has merit but does not fully meet PLOS Global Public Health’s publication criteria as it currently stands. Therefore, we invite you to submit a revised version of the manuscript that addresses the points raised during the review process.

EDITOR:

Thank you for your submission. I have suggested major revisions because TB in Morocco is such an under-reported on topic. However, I note that both the reviewers (privately to me and in their comments to you) have raised significant concerns about your analytic process. In order to proceed toward publication, it is imperative that these are addressed thoroughly. This may include consulting additional experts on spatiotemporal mapping of TB disease for their guidance.

We look forward to receiving your revised manuscript.

Kind regards,

Graeme Hoddinott, Ph.D

Academic Editor

Journal Requirements:

Additional Editor Comments (if provided):

Reviewers' comments:

Reviewer's Responses to Questions

**Comments to the Author**

1. Does this manuscript meet PLOS Global Public Health’s publication criteria? Is the manuscript technically sound, and do the data support the conclusions? The manuscript must describe methodologically and ethically rigorous research with conclusions that are appropriately drawn based on the data presented.? Is the manuscript technically sound, and do the data support the conclusions? The manuscript must describe methodologically and ethically rigorous research with conclusions that are appropriately drawn based on the data presented.

Reviewer #1: Yes

Reviewer #2: Partly

2. Has the statistical analysis been performed appropriately and rigorously?

Reviewer #1: I don't know

Reviewer #2: No

3. Have the authors made all data underlying the findings in their manuscript fully available (please refer to the Data Availability Statement at the start of the manuscript PDF file)?

The PLOS Data policy requires authors to make all data underlying the findings described in their manuscript fully available without restriction, with rare exception. The data should be provided as part of the manuscript or its supporting information, or deposited to a public repository. For example, in addition to summary statistics, the data points behind means, medians and variance measures should be available. If there are restrictions on publicly sharing data—e.g. participant privacy or use of data from a third party—those must be specified.requires authors to make all data underlying the findings described in their manuscript fully available without restriction, with rare exception. The data should be provided as part of the manuscript or its supporting information, or deposited to a public repository. For example, in addition to summary statistics, the data points behind means, medians and variance measures should be available. If there are restrictions on publicly sharing data—e.g. participant privacy or use of data from a third party—those must be specified.

Reviewer #1: Yes

Reviewer #2: Yes

4. Is the manuscript presented in an intelligible fashion and written in standard English?

Reviewer #1: Yes

Reviewer #2: No

Reviewer #1: This is an important area of research in TB and the data from the national programme has been well utilised. Though the study does not add much to the existing knowledge, it documents the process of shifting epidemiology. Some additional details however would make the interpretation of the results better.

1. Whether active case finding was performed in the years for which the data has been used? If yes, what were the approaches, strategies and methods? The number of cases diagnosed and the areas they belonged to would depend on the populations and groups screened.

2. Were there other strategies introduced into the programme during the study period, which might have an impact on the incidence of TB? Many countries had a change from alternate to daily regimen, which led to reduction in recurrent TB.

3. What proportion of the TB patients were retreatment cases?

4. How many patients had missing information and therefore were excluded from the analysis?

Reviewer #2: Tuberculosis remains a threat in the Africa continent with little knowledge on its spatiotemporal patterns; therefore, it is an interesting area to look at. However, I appreciate the authors for their effort to discover the happening around this problem in Morocco. I also thank them for the opportunity to review their manuscript, as I enjoyed reading it. Besides, I have some comments to make, which are below.

Comments

Title:

This should be refine to Spatiotemporal Patterns of Tuberculosis in Urban Slums and Urban–Rural Transition Zones: Evidence from Tétouan, Morocco (2019—2023)

1. Abstract

The results reported are unclear. For instance, “the average incidence was 103 per 100,000”. The question here is average incidence of what?

2. Method

The authors stated in lines 131–132 that additional analysis includes “Regression (binomial and multinomial), comparison (Student’s t-test), and correlation (Pearson’s r) tests were conducted using SPSS Statistics v26.” What was the outcome variable? How did the authors define the outcome variable in the multinomial analysis? In all, none of these particularly multinomial analyses were captured throughout the manuscript.

3. Results

Although authors should stick to reporting percentages for descriptive statistics, the findings reported do not correspond with the figures provided in Table 1. For instance, the authors stated in line 147 that “the median age was 40”, but in Table 1 of Line 389, it was rather the mean age. Also, in Line 147 the ratio of male-to-female is not “1.69” as reported by the authors but 1.70 (male=2272 and female=1338). Besides, in lines 150–151, “Five patients (0.1%) were homeless, and 42 (1.2%) were diagnosed in correctional facilities.” This is not corresponding with evidence provided in Table 1. In Line 389 of Table 1, the number of homeless is 4, and instead of patients diagnosed in prison, the authors reported correctional facilities. Percentages in Table 1 are not also adding up to 100%, even though some are reasonable. The column names of Table 1 are not understandable, with the figures provided for age groups being confusing. Notwithstanding, the sample size (n) of EPTB does not match with that of EPTB Localisations in Table 1.

In line 392, figures reported in Table 2 are also unclear, as a comma is used instead of probably a dot to indicate decimals. Also, analysing the Moran’s I of a pooled 2019—2023 is redundant.

4. General

The analytical approaches stated in the method without a given output are questionable. Again, the inconsistency of findings in the manuscript is regarded as a major issue and warrants reanalysing the data for consideration in other journals.

**Do you want your identity to be public for this peer review?** For information about this choice, including consent withdrawal, please see our Privacy Policy..

Reviewer #1: No

Reviewer #2: No

---

## [Decision Letter · Decision Letter 1]

17 Feb 2026

PGPH-D-25-02866R1

Spatiotemporal Patterns of Tuberculosis in Urban Slums and Urban–Rural Transition Zones: Evidence from Tétouan, Morocco (2019–2023)

Dear Dr. EZ-ZARI,

Thank you for submitting your manuscript to PLOS Global Public Health. After careful consideration, we feel that it has merit but does not fully meet PLOS Global Public Health’s publication criteria as it currently stands. Therefore, we invite you to submit a revised version of the manuscript that addresses the points raised during the review process.

EDITOR:

The reviewers have made sensible suggestions on how to further strebgthen your manuscript.

We look forward to receiving your revised manuscript.

Kind regards,

Graeme Hoddinott, Ph.D

Academic Editor

Journal Requirements:

Additional Editor Comments (if provided):

Reviewers' comments:

Reviewer's Responses to Questions

**Comments to the Author**

Reviewer #3: (No Response)

Reviewer #4: All comments have been addressed

publication criteria? Is the manuscript technically sound, and do the data support the conclusions? The manuscript must describe methodologically and ethically rigorous research with conclusions that are appropriately drawn based on the data presented.? Is the manuscript technically sound, and do the data support the conclusions? The manuscript must describe methodologically and ethically rigorous research with conclusions that are appropriately drawn based on the data presented.

Reviewer #3: Partly

Reviewer #4: Yes

3. Has the statistical analysis been performed appropriately and rigorously?

Reviewer #3: Yes

Reviewer #4: Yes

4. Have the authors made all data underlying the findings in their manuscript fully available (please refer to the Data Availability Statement at the start of the manuscript PDF file)?

The PLOS Data policy requires authors to make all data underlying the findings described in their manuscript fully available without restriction, with rare exception. The data should be provided as part of the manuscript or its supporting information, or deposited to a public repository. For example, in addition to summary statistics, the data points behind means, medians and variance measures should be available. If there are restrictions on publicly sharing data—e.g. participant privacy or use of data from a third party—those must be specified.requires authors to make all data underlying the findings described in their manuscript fully available without restriction, with rare exception. The data should be provided as part of the manuscript or its supporting information, or deposited to a public repository. For example, in addition to summary statistics, the data points behind means, medians and variance measures should be available. If there are restrictions on publicly sharing data—e.g. participant privacy or use of data from a third party—those must be specified.

Reviewer #3: Yes

Reviewer #4: Yes

5. Is the manuscript presented in an intelligible fashion and written in standard English?

Reviewer #3: Yes

Reviewer #4: Yes

Reviewer #3: This is a nice paper describing the spatial-temporal epidemiology of TB in specific part of Morocco. It appears that the authors have responded thoroughly to previous reviewer comments. I have a few suggestions and questions.

Abstract:

1. There are some relative risks provided in the abstract but they need to be accompanied by confidence intervals so readers understand the accuracy. Also, the first time an RR is presented it says “1.90-2.03” but it is unclear if that is a range and if so what type of range. The third instance, it says “RR up to 3.00”, this is unclear, a single RR should be provided with a confidence interval.

Methods:

2. A little more clarity on the spatial units of analysis is needed. The description says there are two urban municipalities and 20 rural districts and the urban area is divided into 18 districts. Is the unit of spatial analysis district/commune, i.e. a total of 38 areas? If so, this should be stated clearly (or the correct statement given if this is incorrect). Also, how many districts are in the old medina (slum Ilke) area? Lower down it says that data were extracted and aggregated – aggregated in what way? To the spatial unit of analysis? Finally, how was urban defined?

3. In the spatio-temporal scan section it says “Relative risk (RR) was calculated for each cluster.” It is unclear what this relative risk is. What was the outcome and what is the risk relative to (i.e. what is the comparison group)?

Table 2:

4. I recommend using fewer decimal places here? Maybe two would be best. Otherwise, it is hard to read this table with so many that are unnecessary.

Results:

5. Several RRs are presented but they need to be accompanied by confidence intervals.

Table 3:

6. Again, confidence intervals need to be presented. These should replace the LLRs, which are not very meaningful to readers.

Discussion:

7. Sentence in lines 307-9 starting “Patterns of population movement…” should have references to support these suggested movement patterns.

8. Although the authors mention the limitation of COVID-19 related under-reporting, they don’t mention general under-reporting which could affect their conclusions more broadly. By which I mean the fact that they are presenting notification data and therefore, low incidence might reflect under reporting rather than genuinely low incidence and similarly, high incidence might reflect more thorough case-finding and identification of cases, which would be a positive thing.

Figure 1:

9. The legend uses the word “evolution” but that is not what is being shown in this figure, it is just a stratification of the temporal trends by clinical form.

Reviewer #4: Uploaded comments

**Do you want your identity to be public for this peer review?** For information about this choice, including consent withdrawal, please see our Privacy Policy..

Reviewer #3: No

Reviewer #4: No

---

## [Decision Letter · Decision Letter 2]

31 Mar 2026

Spatiotemporal patterns of tuberculosis in urban slums and urban–rural transition zones: evidence from Tétouan, Morocco, 2019–2023

PGPH-D-25-02866R2

Dear Mr. EZ-ZARI,

We are pleased to inform you that your manuscript 'Spatiotemporal patterns of tuberculosis in urban slums and urban–rural transition zones: evidence from Tétouan, Morocco, 2019–2023' has been provisionally accepted for publication in PLOS Global Public Health.

Best regards,

Graeme Hoddinott, Ph.D

Academic Editor

Reviewer Comments (if any, and for reference):

Reviewer's Responses to Questions

**Comments to the Author**

Reviewer #4: All comments have been addressed

publication criteria? Is the manuscript technically sound, and do the data support the conclusions? The manuscript must describe methodologically and ethically rigorous research with conclusions that are appropriately drawn based on the data presented.? Is the manuscript technically sound, and do the data support the conclusions? The manuscript must describe methodologically and ethically rigorous research with conclusions that are appropriately drawn based on the data presented.

Reviewer #4: Yes

3. Has the statistical analysis been performed appropriately and rigorously?

Reviewer #4: Yes

4. Have the authors made all data underlying the findings in their manuscript fully available (please refer to the Data Availability Statement at the start of the manuscript PDF file)?

The PLOS Data policy requires authors to make all data underlying the findings described in their manuscript fully available without restriction, with rare exception. The data should be provided as part of the manuscript or its supporting information, or deposited to a public repository. For example, in addition to summary statistics, the data points behind means, medians and variance measures should be available. If there are restrictions on publicly sharing data—e.g. participant privacy or use of data from a third party—those must be specified.requires authors to make all data underlying the findings described in their manuscript fully available without restriction, with rare exception. The data should be provided as part of the manuscript or its supporting information, or deposited to a public repository. For example, in addition to summary statistics, the data points behind means, medians and variance measures should be available. If there are restrictions on publicly sharing data—e.g. participant privacy or use of data from a third party—those must be specified.

Reviewer #4: Yes

5. Is the manuscript presented in an intelligible fashion and written in standard English?

Reviewer #4: Yes

Reviewer #4: The authors have responded to all questions.

**Do you want your identity to be public for this peer review?** For information about this choice, including consent withdrawal, please see our Privacy Policy..

Reviewer #4: No
